# Sex, age, and parental harmonic convergence behavior affect the immune performance of *Aedes aegypti* offspring

Christine M. Reitmayer [1,2,3], Ashutosh K. Pathak[1,2], Laura C. Harrington [4,5], Melinda A. Brindley[1,6,7], Lauren J. Cator [8] & Courtney C. Murdock [1,2,4,5,7,9,10,11✉]

Harmonic convergence is a potential cue, female mosquitoes use to choose male mates. However, very little is known about the benefits this choice confers to offspring performance. Using *Aedes aegypti* (an important vector of human disease), we investigated whether offspring of converging parental pairs showed differences in immune competence compared to offspring derived from non-converging parental pairs. Here we show that harmonic convergence, along with several other interacting factors (sex, age, reproductive, and physiological status), significantly shaped offspring immune responses (melanization and response to a bacterial challenge). Harmonic convergence had a stronger effect on the immune response of male offspring than on female offspring. Further, female offspring from converging parental pairs disseminated dengue virus more quickly than offspring derived from non-converging parental pairs. Our results provide insight into a wide range of selective pressures shaping mosquito immune function and could have important implications for disease transmission and control.

[1] Department of Infectious Diseases, University of Georgia, Athens, GA, USA. [2] Center for Tropical and Global Emerging Diseases, University of Georgia, Athens, GA, USA. [3] The Pirbright Institute, Pirbright Surrey, UK. [4] Department of Entomology, Cornell University, College of Agriculture and Life Sciences, Ithaca, NY, USA. [5] Northeast Center for Excellence for Vector-borne Disease Research, Ithaca, NY, USA. [6] Department of Population Health, College of Veterinary Medicine, University of Georgia, Athens, GA, USA. [7] Center for Vaccines and Immunology, College of Veterinary Medicine, University of Georgia, Athens, GA, USA. [8] Department of Life Sciences, Imperial College London, Ascot, UK. [9] Odum School of Ecology, University of Georgia, Athens, GA, USA. [10] Center for Ecology of Infectious Diseases, Odum School of Ecology, University of Georgia, Athens, GA, USA. [11] Riverbasin Center, Odum School of Ecology, University of Georgia, Athens, GA, USA. ✉email: ccm256@cornell.edu

Vector-borne infections impose a serious health risk to humans—over half of the world's population is at risk for mosquito-borne diseases[1]. An estimated 3.9 billion people, in 128 countries are at risk of dengue virus (DENV) infection, leading to ~390 million dengue virus (DENV) infections per year[2,3]. Further, many diseases that have recently emerged or are re-emerging globally are mosquito-borne (e.g. chikungunya, Zika, and Yellow fever). For example, despite decades of intervention efforts, malaria infections still cause about half a million deaths per year, especially in children under five years of age[4]. The most substantial gains in the fight against these diseases has been primarily through vector control and preventing transmission from mosquitoes to human hosts. However, these efforts are being threatened by the evolution of resistance (both chemical and behavioral) to commonly used insecticides[5–7]. Thus, considerable effort has been made to explore novel ways of controlling mosquito vectors and thereby limiting the dependence on insecticides[8–11].

Many novel insect control technologies involve the mass release of sterile males or males genetically engineered to pass on traits that confer either severe fitness costs (i.e., population suppression approaches[8,12–14]) or enhanced resistance to human pathogens (i.e., population replacement approaches[15–17]) to offspring. Ensuring that released males are able to effectively compete for, and mate with, wild-type females is a key component to the success and sustainability of these programs. Yet, many key aspects of mating interactions such as the role of female mate choice and the benefits females gain from being choosy remain unclear. Thus, models built to predict successful release strategies often do not account for variation in male mating success[18–21].

Several important mosquito vector species mate in aerial swarms[22–24], where males use a combination of visual and chemical cues in the selection of sites for swarming[24–28]. In the case of Ae. aegypti, the host is the site for mating encounters. Females approach the host or swarm individually, and copulas from mid-air and are highly transient (9–31 s)[22]. Consequently, females are represented at much lower numbers than males in the swarm, resulting in a male-dominated operational sex ratio[25,29]. The surplus of males decreases the likelihood that any particular male will mate, and the majority of males within the swarm are believed to go unmated. Further, observations suggest that female mate choice is important in determining male mating success, with females displaying a variety of active rejection behaviors (e.g., abdominal tilts, tarsal kicks, thrusts, and holds) toward undesired males[30–34] resulting in a consistently low proportion of successful copulations when individual mating encounters are observed[30,32,33]. Finally, male mosquitoes are strongly attracted to the sounds produced by the wingbeat of a female mosquito, allowing them to locate prospective mates in the aerial swarm[35–39]. As the male approaches the female, both sexes engage in a dynamic acoustic interaction where both individuals modulate their wingbeat frequency so it overlaps at a harmonic overtone[40–43]. This phenomenon, harmonic convergence, has been suggested as a mechanism used by females to assess and select among potential mates[30,41], but see refs. [44,45].

Despite this current understanding, there is limited knowledge on what fitness advantages females might gain from being selective. In mosquitoes, the fitness benefits females gain (by mating with males during successful harmonic convergence events) described to date have been both direct and indirect. A recent study in Ae. aegypti demonstrates that male seminal proteins increase female survival suggesting that mating with a certain male could potentially increase female fitness directly.[46] Additionally, the sons of pairs that harmonically converged were more likely themselves to harmonically converge in mating interactions and in turn experienced higher mating success[30]. An extension of sexual selection theory that could have important

ramifications for the success of novel vector control technologies utilizing genetically modified males (and vector-borne disease transmission in general) involves the selective role of parasites and pathogens in mediating mate choice. Parasites and pathogens impose strong selective pressures on their hosts and are capable of regulating host populations[47]. The immune system is an organism's primary defense against parasitic/pathogenic invasion. The theory of parasite-mediated sexual selection predicts that females use proximate cues (i.e., mating displays, ornamentation, etc.) as a signal of the potential mate's ability to resist parasites and pathogens, which can then benefit her directly (by minimizing her contact with any parasites and pathogens the male might carry) or indirectly when these resistance traits are passed on to her offspring[48].

In this study, we explore the relationship between harmonic convergence and offspring immune performance in the yellow fever mosquito, Aedes aegypti. First, we assessed the performance of mosquitoes with melanization and bacterial challenge assays. These immune challenges were chosen because they comprise both cellular and humoral immune responses that are evolutionarily conserved, are important for a wide diversity of pathogens/parasites, and have known fitness costs[49,50]. Additionally, we explored the ramifications of harmonic convergence on metrics of vector competence (i.e., the proportion of mosquitoes infected, disseminated, and infectious) for an important human pathogen, the flavivirus DENV (serotype DENV 2). Finally, energy invested into immune defense cannot be allocated to other life-history processes (e.g. growth, mating, producing eggs, etc.), and organisms must balance energy allocation across these competing life-history demands in order to maximize fitness[51]. Thus, investment in immunity can differ between the sexes[52], with age (immune senescence)[53–55], and with investment in mating[56,57] or parental effort.[58] In light of this, we assayed offspring immune performance based on (1) harmonic convergence status, (2) sex, (3) age, (4) physiological stage (non-blood-fed vs. blood-fed), and (5) mating status (Fig. S1).

## Results

Depending on the immune assay (melanization, in vivo bacterial growth) and experimental group, we assessed the immune performance in 3–5 female and male offspring from 6 to 10 parental pairs that had either converged or not converged (total number of observations shown in Tables S1, S3, S5, S6, and S8). A total number of 804, 871, and 673 mosquitoes were assessed for melanization ability, mortality in response to bacterial challenge, and in vivo bacterial growth, respectively. A total of 4263 eggs were generated from four recording events for DENV infections, divided according to parental convergence status ($n = 2123$ from converged and $n = 2140$ from not converged parental pairs), and randomly allocated across three replicates. A total of 720 tissue samples were collected for infection (body), dissemination (head and legs), and infectiousness (saliva). All body samples were analyzed for infection ($n = 720$), with heads/legs ($n = 448$) analyzed for dissemination from females with DENV positive bodies, and saliva ($n = 206$) analyzed for infectiousness from females with DENV positive bodies and heads/legs.

**Humoral melanization.** Females exhibited a more robust melanization response than males (Fig. 1a, Tables S1 and S2). After 24 h of in vivo incubation, 80% of the beads recovered from females were either fully melanized (41%) or partially melanized (39%), and only 20% of recovered beads were unmelanized. In contrast, almost half of the beads recovered from male mosquitoes after 24 h were unmelanized (47%), followed by partially melanized (30%) and fully melanized beads (23%) (Table S2). Parental convergence

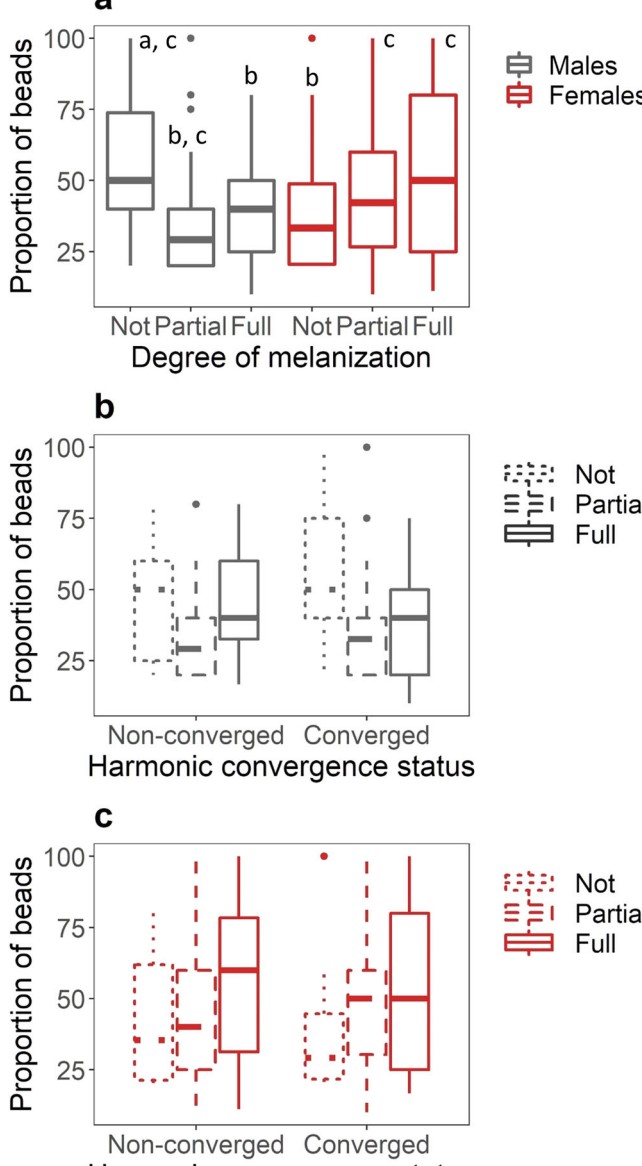

**Fig. 1 The effect of sex and harmonic convergence on melanization ability.** The mean proportion of beads that were unmelanized (Not), partially melanized (Partial), or fully melanized (Full) 24 h after individual *Aedes aegypti* were injected with a single, neutrally charged G25 Sephadex bead and housed at 27 °C ($n = 641$ observations). Boxes are boxplots of the interquartile range with medians for each treatment group, whiskers represent the first and third quartiles, and points are outliers in the data. Male and female mosquitoes are colored gray and red, respectively. **a** Main effect of sex on melanization ability. Letters denote significant differences among treatment groups ($p < 0.05$) determined by Tukey's contrasts. **b** The effect of parental convergence status (non-converged vs. converged) on melanization ability in male mosquitoes. **c** The effect of parental convergence status on melanization ability in female mosquitoes. In both **b** and **c** unmelanized, partially melanized, and fully melanized beads are represented by dotted, hashed, and solid lines, respectively.

status affected the melanization response of male offspring (Fig. 1b, Tables S1 and S2) but not female offspring (Fig. 1c, Table S1 and S2). Males derived from parental pairs that had successfully converged had a lower probability of melanizing beads (51% unmelanized) relative to males from parents that did not harmonically converge (42% unmelanized) (Table S2). In

female offspring, we did not observe a significant effect of parental convergence status on melanization response. We also did not observe a significant effect of mating or blood-feeding status on melanization response (Table S1).

**Bacterial growth and mosquito mortality**. We observed significant effects of parental convergence status, as well as offspring sex, age, and blood feeding status (measured for 5-day-old females only) on bacterial growth within the mosquito (Tables S3 and S4). Overall, reduced bacterial growth was observed in females relative to male offspring, younger relative to older offspring, and offspring from converged relative to non-converged parental pairs. Out of all experimental groups examined, only blood-fed females were able to reduce bacteria numbers below the level of initial bacteria inoculation (200,000 *E. coli*).

In addition to these main effects, the effects of age and parental convergence varied between the sexes. In both sexes, offspring from parents that had successfully converged were more capable of limiting bacterial growth at an early age compared to offspring from non-converging parents (Tables S3 and S4). However, this effect was significantly more pronounced in young, male offspring (age × sex × parental convergence status interaction, Fig. 2a, Tables S3 and S4). Within females, mated 3-day-old females were less able to control bacterial growth relative to unmated 3-day-old females. In contrast, there was little difference in male resistance to bacterial infection with mating status (Fig. 2b, Tables S3 and S4). Finally, a marginally significant interaction between age × sex on the effect of bacterial infection on mosquito mortality 24 h post-infection (Fig. 2c, Table S5) indicates that overall mortality associated with bacterial infection increases with age more so for male than female offspring.

**DENV infections**. Overall, we did not observe a significant effect of parental convergence status on the proportion of mosquitoes that became infected, showed disseminated infection, or that became infectious with DENV (Table S6, Fig. 3). There was no significant interaction between infection status (analyzed by individual tissue type), time post-infection, and convergence status (Table S6). However, parental convergence status affected the rate of infection dynamics. A significantly lower proportion of females derived from converging parental pairs showed detectable infection early on (3 days post-infection) (Tukey's estimate (log-odds) = −0.86, se = 0.34, $p = 0.0126$, Table S7, Fig. 3a) but a significantly greater proportion of these progeny reached levels of detectable disseminated infection earlier compared to offspring from non-converging pairs (12 days post-infection) (Tukey's estimate (log-odds) = 0.62, se = 0.258, $p = 0.0183$, Table S7, Fig. 3b). Offspring from converging pairs seem to have low initial midgut viral replication rates, leading to a lower number of detectable infection cases at the 3 days time point. However, in the same group we then observed a rapid increase in disseminated infection prevalence between 6 and 9 dpi, almost plateauing at 12 dpi, whereas we observed a slower, more gradual increase of disseminated infection prevalence in offspring derived from non-converged parental pairs. We did not observe any significant differences in the proportion of infectious mosquitoes over time between offspring from converged vs. non-converged parents (Table S7 and Fig. 3c). However, it should be noted that the power of statistical analysis of this subset of data is limited by the number of DENV positive saliva samples, which was lower than expected based on initial infection prevalence across the two experimental treatment groups. These results were confirmed by CPE assay and qPCR screening for viral RNA (see Fig. S3). Finally, we did not observe a significant difference in daily survival over the course of the experiment (18 days) between female

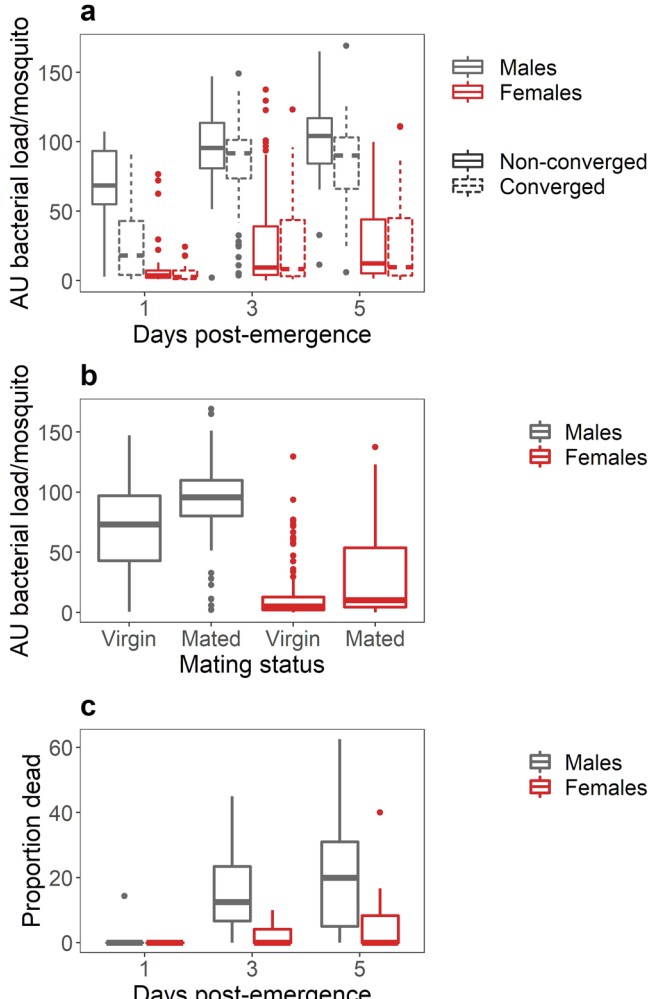

**Fig. 2 The effect of sex, age, and harmonic convergence on bacterial growth and mosquito mortality.** Bacterial growth of *Escherichia coli* after 24 h in vivo incubation in *Aedes aegypti* mosquitoes injected with 200,000 live bacteria and housed at 27 °C ($n = 577$). Boxes are boxplots of the interquartile range with medians for each treatment group, whiskers represent the first and third quartiles, and points are outliers in the data. Males and females are represented by the colors gray and red, respectively. **a** The effect of sex, age, and parental convergence status on bacterial growth. Solid and dotted lines represent mosquito offspring from parental pairs that did not converge and that did converge, respectively. Significant pairwise comparisons are indicated in Table S4. **b** The effect of mating status (Virgin or unmated vs. Mated) on bacterial growth for males (gray) and females (red) ($n = 305$ observations). **c** The effect of mosquito age (Days Post-Emergence) on the mortality of male (gray) and female (red) mosquitoes 24 h post-challenge with live bacteria ($n = 121$ observations).

offspring derived from converged and non-converged parental pairs (survival of >95%).

To further examine if temporal trends in the proportion of offspring that were infected, had disseminated infection, or were infectious with DENV-2 varied by convergence status, a nested/hierarchical model was specified (Table S8). Similar to the full model above (Table S6), the nested model confirmed that parental convergence status does not affect the overall proportion of mosquitoes infected, with disseminated infection, or infectious with DENV. However, convergence status did appear to influence the temporal dynamics of these response variables within each tissue. Viral prevalence in the bodies of offspring from converging parental pairs increased over time (linear: effect, estimate = 4.58,

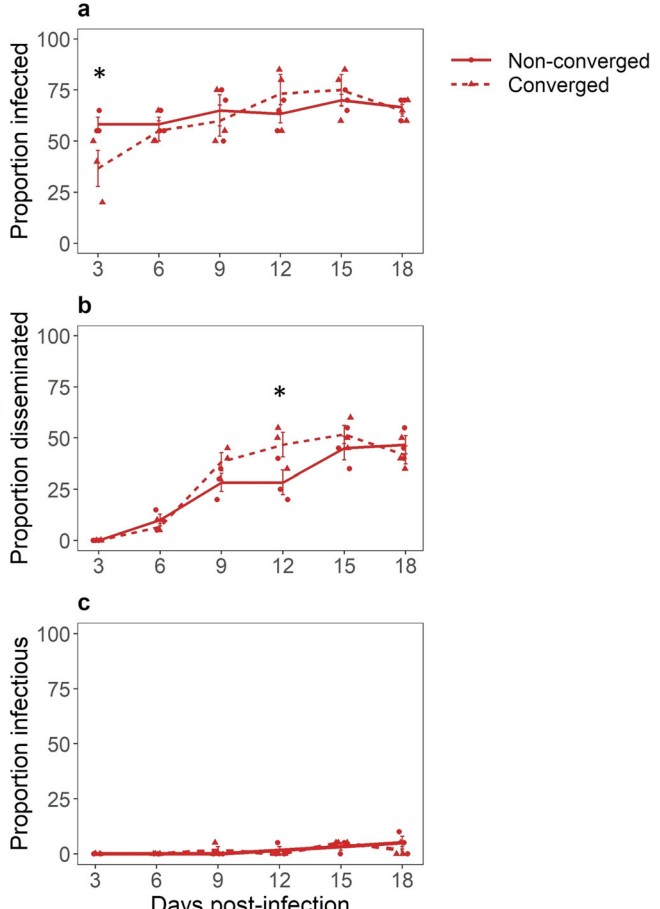

**Fig. 3 The effect of harmonic convergence on dengue-2 virus infection.** The effect of parental convergence status (Non-converged, solid lines; Converged, dotted lines) on different metrics of dengue-2 virus (DENV) infection in female *Aedes aegypti* offspring. **a** Proportion of infected mosquitoes with DENV positive bodies on days 3, 6, 9, 12, 15, and 18 post infection ($n = 720$ observations). **b** Proportion of mosquitoes with disseminated infection, with DENV positive heads and legs on 3, 6, 9, 12, 15, and 18 days post-infection ($n = 448$ observations). **c** Proportion of infectious mosquitoes with DENV positive saliva on 3, 6, 9, 12, 15, and 18 post infection ($n = 206$ observations). Data points represent mean values for each treatment and each biological replicate with error bars showing standard error of the overall mean at each time point post infection. Significant differences in infection and dissemination rate at a certain time point are indicated by asterisks ($p < 0.05$).

se = 1.16, z-value = 3.97, $p = $ <0.001; quadratic: effect, estimate = −3.01, se = 1.16, z-value = −2.59, $p = 0.01$). This increase was due to the proportion of mosquitoes infected at 12, 15, and 18 dpi relative to earlier time points (Fig. 3a). In contrast, the proportion of offspring from non-converging parental pairs did not vary significantly over time (linear: estimate = 1.64, se = 1.14, z-value = 1.44, $p = 0.151$; quadratic: estimate = −0.31, se = 1.14, z-value = −0.27, $p = 0.789$) (Fig. 3a). A similar pattern was observed for viral dissemination: the proportion of offspring from converged parents with DENV positive heads/legs changed over time (linear: estimate = 15.32, se = 2.5, z-value = 6.13, $p = $ <0.001; quadratic: estimate = −9.55, se = 1.97, z-value = −4.86, $p = $ <0.001) as well as from non-converged parents (linear: estimate = 12.85, se = 2.17, z-value = 5.92, $p = $ <0.001; quadratic: estimate = −4.66, se = 1.75, z-value = −2.66, $p = 0.008$). However, the predicted quadratic effect on the proportion of mosquitoes with disseminated infections was stronger for females

from converging parents (illustrated by the stronger initial increase and turn-over of this response variable with days-post infection, Fig. 3b, Table S8) than offspring from parents that did not converge. There were no temporal trends in the proportion of mosquitoes with DENV positive saliva regardless of parental convergence status (Fig. 3c). Taken together, these results indicate that while parental convergence status affects the temporal dynamics of early infection and dissemination, it does not affect the overall average proportion of infected, disseminated, or infectious mosquitoes.

## Discussion

The majority of work conducted on mosquito immune responses over the past 30 years has primarily focused on characterizing mosquito immunity in female mosquitoes of a given age and reproductive stage. In this study, we found that mosquito immune responses were determined by interactions between sex, age, reproductive status, and parental mating behavior. Our results have wide-ranging implications for understanding the selective pressures shaping mosquito immune function. In particular, they highlight the role of trade-offs in resource allocation across multiple competing life history processes, and the impact these relationships have on disease transmission. Further, our work has implications for the development of novel vector control tools.

Parental convergence status differentially affects female and male offspring melanization ability. Melanization is the product of a series of enzymatic and non-enzymatic reactions beginning with the hydroxylation of tyrosine and ending with the oxidate polymerization of indolequinones[59]. Melanization has been implicated in the defense of mosquitoes to a variety of parasites and pathogens (reviewed in ref. [59]). Overall, female mosquitoes were more likely to produce a melanization response to beads relative to males. This result is consistent with a previous study that also showed stronger melanization in female mosquitoes compared to males in response to *Dirofilaria immitis microfilariae*[60].

In males, progeny from parents that harmonically converged demonstrated lower melanization rates than those from parents that did not converge. One potential explanation for this is that male offspring with fathers that successfully converged may allocate resources across energetically competing life history demands (like mating and mounting an immune response[61–63]) differently than males produced by fathers that did not successfully converge. Upon emergence, the fitness of a male mosquito will be dependent upon his ability to compete for access to female mosquitoes in a mating swarm and the number of females he successfully mates with. Both mating and mounting immune responses, including the melanization response[64,65], have been shown to incur significant energetic costs across a diversity of invertebrate and vertebrate systems (reviewed in ref. [66]). Thus, male offspring produced by desirable fathers may invest more resources into mating to maximize lifetime fitness. In contrast, we did not see an effect of mating behavior, mating status, or blood feeding status on the melanization response of females. Taken together, these results highlight the differences in selective pressures experienced by male and female mosquitoes.

The mosquito innate immune system comprises several antimicrobial strategies with hemocytes playing a crucial role in the cellular immune response and antimicrobial peptides, pattern-recognition receptors, and components of the phenoloxidase cascade being key components of the humoral immune response[67–69]. Intra-thoracic injections of *E. coli* have been previously used to investigate the cellular immune response of mosquitoes[70], measure the expression of antimicrobial peptides in mosquitoes[71,72], age-associated mortality after bacterial

challenge[73], and compare immune response mechanisms between mosquito species[74]. Similar to the melanization assay, female mosquitoes exhibited more robust bacterial killing responses and enhanced survival during bacterial infection relative to males of the same age and stage. Surprisingly, bloodfed females were the only treatment group that managed to reduce bacterial numbers within 24 h post-infection. This could be an indirect effect of the increase in oxidative and nitration stress associated with digestion of the bloodmeal[75,76], the induction of immune responses that are triggered to control microbial proliferation of midgut bacteria that occurs in response to the bloodmeal[77–79], or an increase in hemocyte mitosis and the phagocytic response against *E. coli*[80,81]. Additionally, young adult mosquitoes of both sexes were more likely to survive a bacterial infection and to inhibit bacterial growth in vivo than older mosquitoes. This result is not entirely surprising considering significant immunosenescence has been observed in both phagocytosis of live *E. coli* and the number of hemocytes as early as day four post-emergence in *Ae. aegypti*[73].

Immunosenescence was more pronounced in males than females, possibly reflecting the distinct reproductive demands males and females encounter to maximize lifetime fitness. Male reproductive success is solely determined by the number of females he mates with after emergence, while female reproductive success is determined by her mating success but also by the energetically demanding production of eggs throughout her life. Thus, females, who may encounter a higher pathogen load throughout their adult lives due to the necessity of taking blood meals to support egg production, may invest more in immunity across a greater proportion of their lifespan in order to reduce any negative impacts of infection on her lifetime reproductive success. Finally, we observed male offspring from parents that successfully converged, experience a significant bacterial killing advantage early in adulthood relative to male offspring from parents that failed to successfully converge. In contrast to our melanization results, male offspring from converging fathers inherit an enhanced ability to fight a bacterial infection upon challenge during early adulthood, therefore, increasing their probability of reproductive success. Taken together, these results suggest that males from converged parents allocate resources across mating effort and immune defense differentially depending on the nature of the immune challenge and physiological cost.

While sephadex bead challenge and infection with *E. coli* bacteria stimulate different components of the mosquito immune system, we also wanted to assess the effects of harmonic convergence behavior on the ability of female offspring to transmit human pathogens. Thus, we assessed whether or not parental convergence status impacted different DENV infection metrics in female mosquitoes. Although we did not observe a substantial effect of parental convergence status on the overall proportion of female offspring that became infectious, we did observe an effect of parental convergence on the proportion of detectable infection early in the infection process as well as in the DENV-2 dissemination dynamics over the course of infection. Female offspring from parents that successfully converged initially showed fewer detectable infection cases early on (3 days post-infection) relative to female offspring from parents that did not converge. These female offspring from converging parental pairs may differ in arbovirus susceptibility factors early in infection such as those that promote or regulate oxidative stress (e.g. antioxidant enzymes like catalase) leading to a slower viral replication rate in the midgut[82–84]. Perhaps, females from converging parents can tolerate higher oxidative and nitration stress induced by the production of reactive oxygen and nitrogen species with the break-down of heme proteins to fuel egg development while also promoting early resistance to DENV infection as compared to female offspring from non-converging parents.

Interestingly, female offspring from parents that successfully converged, while initially less competent to facilitate viral replication actually exhibited faster DENV dissemination rates. This suggests that DENV replication and escape from the midgut epithelium, as well as infection of secondary tissues, may occur at a higher rate in these females. This could happen for a couple of reasons. Females from converging parental pairs may differ in their overall investment in host defense mechanisms (such as RNAi[85–89], nitration[90], etc.)[91,92], or in their susceptibility to mechanisms DENV utilizes to evade/suppress the host responses[93–96]. We did not see a difference in survival of DENV infected female offspring from parental pairs that converged compared to those that did not converge, suggesting that female offspring from converged parents could be investing less in immunity in favor of future reproduction[57,62,97]. Alternatively, if harmonic convergence is an honest indicator of male genetic quality, then offspring resulting from these pairs may have higher overall fitness than females from parents that have not converged[30]. The intracellular viral replication cycle is highly dependent on cellular lipids and sterols, with DENV infection in cell culture and in vivo relying on intracellular cholesterol levels[98–102]. Thus, if females from converging parents have a better body condition (e.g. higher levels of teneral reserves), they may in fact provide more intracellular resources to support viral replication and production. Future work should address the effects of parental convergence status on DENV specific immune responses, viral replication and dissemination, and virus mechanisms to evade/suppress host defense mechanisms in female offspring.

Overall the proportion of mosquitoes that became infected and disseminated DENV in our experiment are similar to previous work done by our group[103] as well as others[104]. However, we observed a very low proportion of females at 18 days post-infection with DENV in their saliva, which may have inhibited our ability to discern the effect of parental harmonic convergence on the probability of becoming infectious. We confirmed the data acquired in the PFU assays with CPE assays, as well as by measuring viral RNA levels. Several reasons could explain this: First, the *Ae. aegypti* strain used in this study was a low-generation, field-derived outbred strain that may be a less competent vector of DENV overall. Second, this result could be specific to the DENV-2 strain we used or the mosquito–virus combination[105,106]. Third, differences in how arbovirus infection experiments are executed across labs can lead to variations in reported transmission potential estimations (e.g. viral titers of infectious feeds, variation in cell lines used during the culturing process, etc.). Future studies comparing the vector competence of this strain with other *Ae. aegypti* or DENV-2 strains could confirm this and could further clarify the effect of parental mating behavior on DENV-2 susceptibility.

Our results indicate that parental mating behavior can be used as a predictor of offspring immune performance, with immunity being enhanced or reduced depending on the nature of the immune challenge and likely the associated physiological cost. Overall, we found that parental mating behavior influenced immune performance in varying ways driven by offspring age and sex as well as the reproductive status of female progeny. Our study has several implications for the design and efficacy of vector control technologies that rely on releasing transgenic male mosquitoes to mate competitively with wild-type females. First, genetic modification altering mosquito immune function might potentially lead to trade-offs with other life history processes, which in turn affect male offspring differently than females. This is particularly important to consider for strategies in which male offspring are required to mate competitively with wild females to deliver control. If the mating competitiveness of transgenic males is impaired, the transgene might not spread quickly through a population in the field or may be rapidly selected against. Second, if a male's ability to harmonically converge signals his overall quality, there is also the potential for female avoidance of genetically manipulated males if there are substantial fitness costs associated with the introduced modification[107,108]. Finally, the efficacy of the introduced modification will likely vary with factors like mosquito sex, age, and reproductive status as well as environmental context. In light of these results, we suggest that variation and evolutionary determinants of mating competitiveness of transgenic males should be further characterized and incorporated into models predicting the spread of a promising transgenic mosquito strain in the field. Further, we argue that male mating success should be assessed using wild-type field-derived female mosquitoes. Mosquitoes reared under optimal conditions for multiple generations may face reduced sexual selection pressure, and the importance of female mate choice might be lower compared to what released males might encounter in the field.

In general, the selective pressures that shape the evolution of mosquito immunity in nature represent a critical gap in our understanding of mosquito evolutionary ecology. Specifically, we need to address the importance of trade-offs between investment in immunity and traits that impact mosquito fitness and pathogen transmission. Further, in order to predict evolutionary responses, a better understanding of how environmental variation is predicted to shape the nature and magnitude of these trade-offs is needed. Recent work in *Anopheles* mosquitoes provided evidence that sexually selected traits affect immunity and vectorial capacity[109,110]. Our results reinforce the findings from these and other studies conducted across a wide diversity of vertebrate and invertebrate taxa, showing that immune response induction and efficiency differ between the sexes[52], decline with age[53–55], and vary with investment in reproduction (mating and parental effort)[56,57]. Our results also suggest that male and female immune responses could have evolved in response to different selective pressures, with females investing more in immunity overall, with the male immune response affected by age, parental convergence status, and likely the underlying physiological cost. Finally, future studies are warranted to understand how genetic variation across mosquito populations, as well as environmental variation, contributes to selection on mosquito immune responses, variation in female mate choice, and the ability to harmonically converge. The effect of mate selection on offspring fitness may vary by the specific traits assessed[111] and will likely depend on environmental context. Together, the results of this research have broad application to the fields of evolutionary biology, mosquito–pathogen interactions, and the development of novel mosquito control technologies to mitigate the transmission of mosquito-borne pathogens.

## Methods

**Mosquito rearing.** Eggs (F2–F4, originating from Kamphaeng Phet Province, Thailand (16°27′ 48″N, 99°31′ 47″E) were hatched under a reduced pressure vacuum desiccator for 30 min in ddH$_2$O and provided with 0.3 mg of crushed Cichlid pellets (Hikari Cichlid Gold Baby Pellets). After 24 h, larvae were dispensed into larval trays at a density of 200 larvae per tray, with each tray containing 1 L of distilled water and four Cichlid pellets (Hikari Cichlid Gold Large Pellets). After 7–8 days, pupae were collected, and adults were allowed to enclose into large mating cages. We maintained adults on 10% sucrose solution ad libitum. To generate eggs, sugar water was removed for 48 h and water removed for 24 h. Females were then offered a whole human blood meal (Interstate Blood Bank, Memphis, TN, USA) in an artificial glass feeder with a bovine intestine membrane. Eggs were collected on paper towels in a damp oviposition site 3–4 days after the blood meal. Eggs were allowed to embryonate for a minimum of 1 week and stored dry on paper in humidified chambers. Mosquito colonies were maintained at a constant 27 ± 0.5 °C and 85 ± 5% relative humidity in environmental insect chambers (Percival Scientific) set to a 12 h light:12 h dark photoregime.

Experimental mosquitoes were reared as above except for two differences: application of moderate stress and lower larval density. Previously, we found that

in order to detect differences between offspring of converging and non-converging males, we needed to apply a moderate level of stress[30]. Fourth instar larvae were exposed to $20 \pm 0.5\,°C$ for 24 h, as moderate stress conditions[30]. Secondly, larval rearing density was lower (usually between 80 and 150 larvae per tray) as eggs from each female were hatched separately and the total amount of food was adapted accordingly. Pupae were separated according to sex and allowed to eclose in sex-specific cages.

**Harmonic convergence assays.** We recorded acoustic interactions between opposite-sex pairs of unmated 3–5-day-old *Ae. aegypti*. Females were anesthetized on wet ice and tethered to a 2 cm strand of human hair using Nailene glue (Pacific World Corp., San Diego, CA, USA). The hair was attached to a piece of metal wire as described in Cator et al. (2009)[41]. One female per trial was positioned in the center of a $15 \times 15 \times 15$ cm Plexiglas mating arena, 2 cm above the sensitive face of a particle velocity microphone (NR-21358; Knowles, Itasca, IL). The arena was placed on a heat plate set to 30 °C, and a tray with moist cotton wool was provided to increase humidity in the arena. The female flight was initiated via the removal of tarsal inhibition and gentle puffs of air. One male per trial was released into the mating arena and allowed to mate with the female.

Acoustic recordings were taken from the time of male release to the end of any type of first physical interaction between male and female or the termination of successful copulation. Trials in which males took longer than 5 min to initiate an interaction with the female were discarded. The acoustic interactions of mosquito flight tone were recorded and analyzed using Raven 1.0 software (Cornell Laboratory of Ornithology, Ithaca, NY). Harmonic convergence was defined as when the harmonic frequencies of both the male and female during an encounter matched. Frequencies were considered to be matching if they were within less than 4.95 Hz of each other and lasted in this state for a minimum of 1 s[30,41].

Regardless of whether or not harmonic convergence occurred within a mating interaction, male and female pairs were maintained together in 500 ml paper cups (supplied with 10% sucrose and kept at $27 \pm 0.5\,°C$ and $85 \pm 5\%$ relative humidity) to achieve a high number of successfully mated pairs. The male was then removed after 3 days. Females were blood-fed as described above, and eggs were collected from each female from up to three separate gonotrophic cycles and stored as described above.

**Immune assays.** To stimulate humoral melanization and bacterial killing, immune challenges were administered to mosquitoes anesthetized on wet ice via an intra-thoracic injection into the anepisternal cleft[112] using a microcapillary glass needle (P-97, Sutter Instruments, Novato, CA, USA) attached to a mouth pipette (humoral melanization) or a Nanoject injector (Drummond Scientific, Broomall, PA, USA), for bacterial growth[71].

*Humoral melanization assay.* To stimulate the melanization response, mosquitoes in each experimental group were injected with one neutrally charged G-25 Sephadex bead (Sigma, St. Louis, MO, USA)[113]. As Sephadex beads range in size from 40 to 120 μm diameter, we visually selected the smallest beads for inoculation. Beads were injected using a minimal amount of Schneider's *Drosophila* medium (less than 0.5 μl). After the injection, mosquitoes were housed individually in 50 ml tubes and maintained as described above. After 24 h, mosquitoes that were able to walk were anesthetized on ice and beads were dissected out in a phosphate-buffered saline (PBS) solution. Recovered beads were scored according to their degree of melanization using three categorical classes: unmelanized, partially melanized, or fully melanized[71,114–116].

*Bacterial growth and mosquito mortality.* Tetracycline-resistant GFP-expressing *Escherichia coli* (*E. coli*), dh5 alpha strain, were grown overnight in Luria-Bertani's (LB) rich nutrient medium in a shaking incubator at 37 °C. We inoculated ice anesthetized mosquitoes in each experimental group with 200,000 live *E. coli* bacteria suspended in Schneider's *Drosophila* medium (Gibco) at a final volume of 69 nl using intra-thoracic injections.

After injection, mosquitoes were housed individually in 50 ml Falcon tubes and maintained as described above. After 24 h, the number of dead mosquitoes was recorded to estimate the effects of experimental treatment on bacterial virulence. Mosquitoes that survived the 24 h post-challenge, were homogenized in 1 ml of LB medium and the suspension transferred into culture tubes containing 1 ml of LB medium. The bacteria solution was incubated for 5 h in a shaking incubator (shaking speed 180 rpm) at 37 °C. Afterward, 200 μl of each sample were pipetted into black 96-well plates in duplicate, and GFP fluorescent signal was measured using a multimode microplate reader (Varioskan, Thermo Scientific, Waltham, MA, USA). Fluorescence values for the initial 200,000 *E. coli* bacteria injection dose were obtained and used as a baseline to normalize experimental values. Methods associated with validating this protocol are provided in the supplementary information (SI Methods).

**Dengue infections**
*Experimental design.* Eggs from parental pairs that had either harmonically converged or did not converge were collected as described above. Eggs from four recording events (from a total of 55 parental pairs) were separated according to

parental convergence status (not converged vs. converged) and divided into three replicates. Eggs from each treatment and replicate were hatched separately, as described above. For each experimental treatment and replicate, pupae were transferred into mixed-sex cages to allow mating, and adults were maintained as described above. Four to five days after emergence, males were removed, and sucrose was replaced with $dH_2O$. The next day, 5–6-day-old females were provided a DENV infectious blood meal.

*DENV in vitro culturing and mosquito infections.* We propagated DENV-2 virus stocks (strain PRS225488), originally isolated from human serum in Thailand in 1974 and acquired from the World Reference Center for Emerging Viruses and Arboviruses at the University of Texas Medical Branch, using previously described methods[103]. The blood meal consisted of 50% (vol/vol) human red blood cells (washed three times with RPMI and cleared from white blood cells and residual plasma), 33% (vol/vol) Dulbecco's Modification of Eagle's Medium (DMEM, Corning) containing DENV at a final concentration of $2.8 \times 10^5$ (replicate 1), $1.5 \times 10^5$ (replicate 2), and $2.3 \times 10^5$ (replicate 3) PFU/ml (titrated from a sample of the prepared blood meal and determined using the Spearman–Karber $TCID_{50}$ method[117]), 20% (vol/vol) FBS, 1% (wt/vol) sucrose, and 5 mM ATP. The blood meal was provided through a glass feeder as described above. Afterward, blood-engorged females were randomly distributed into 500 ml paper cups with 23–28 females each. Adults were maintained in cups as described above.

*Determination of infection and dissemination.* Dengue infection in mosquitoes can be assessed through three key stages: infection, dissemination, and infectiousness. Per parental convergence group and replicate, 20 mosquitoes were processed on days 3, 6, 9, 12, 15, and 18 post-infection ($n = 120$ females per time point, $n = 760$ females in total). To do this, females were immobilized on ice and then transferred to a chill table (Bioquip, Rancho Dominguez, CA). Legs and wings were removed, and the proboscis inserted into a 200 μl pipette tip with the end cut-off and containing 30 μl of salvation mix (1900 μl FBS, 20 μl 300 mM ATP, 80 μl red food dye). Legs were kept for analysis of dissemination and transferred into 2 ml tubes containing DMEM with 1× antibiotic/antimycotic. Females were allowed to salivate for 40 min on a heat plate. Successful salivation was confirmed by the presence/absence of red food dye in female abdomens. Afterward, the saliva was transferred into Eppendorf tubes containing 700 μl of DMEM with 1× antibiotic/antimycotic to test for infectiousness. Heads of females were then cut off and added to the tubes containing the leg samples to test for dissemination. The remaining body was transferred into Eppendorf tubes containing 700 μl DMEM with 1× antibiotic/antimycotic to test for infection.

*DENV assays.* Using plaque assays as described previously[103,118], we tested for the presence of viable DENV particles in the body, head, and legs, and saliva as a proxy for mosquitoes that are infected, have disseminated infection, or are infectious, respectively[119]. All body samples taken were analyzed for infection. Those animals that tested positive for infection, subsequently had head and leg samples tested for DENV dissemination. Finally, those that successfully disseminated virus had saliva tested for infectiousness. Body samples, and head and leg samples, were homogenized using a QIAGEN TissueLyzer at 30 cycles/s for 30 s and centrifuged at $17,000 \times g$ for 5 min at 4 °C. After centrifugation, the supernatant was used to inoculate Vero cells. After an initial infection period of 2 h, media was removed and an overlay 1.5% UltraPure low melting point agarose (Invitrogen)/DMEM with 1× antibiotic/antimycotic was added. Samples were subsequently incubated at 37 °C, 5% $CO_2$ for 6 days. After incubation cells were fixed with 10% formalin and stained with 1% crystal violet.

Cypopathic effect assays (CPE) were performed like described above for plaque assays, however, cells were scored for whole well cytopathic effect instead of individual plaques.

Molecular detection of dengue RNA was performed with the DENV-specific primers D1 (5'-TCAATATGCTGAAACGCGCGAGAAACCG-3') and D2 (5'-TTGCACCAACAGTCAATGTCTTCAGGTTC-3') producing a 511 bp fragment[120]. Total RNA was extracted using the PureLink RNA extraction kit as per the manufacturer's instructions (Thermo Fisher Scientific, Waltham, MA, USA). Reverse transcription and PCR reactions were performed in a single-step procedure with iTaq Universal Probes One-Step Kit as recommended by the manufacturer (Bio-Rad, Hercules, CA). Reaction conditions were first optimized using cell culture supernatants with a known starting PFU of $10^7$/ml. Supernatants were serially diluted 10-fold with the final dilution representing 1 PFU/ml before RNA extraction and single-step RT-PCR. PCR product was assessed with 1% agarose gel electrophoresis.

**Statistics and reproducibility.** Statistical analyses were performed within the RStudio-integrated development environment for R.[121,122] To assess the effects of convergence status (*non-converged* vs. *converged*), sex (*male* vs. *female*), age (*1* vs. *3* vs. *5* days old), and reproductive stage (*unmated* vs. *mated* or *nonbloodfed* vs. *bloodfed*) on mosquito immune performance, we ran three model analyses with the following fixed effects. *Model 1* investigated the effects of (i) convergence status, (ii) sex, and (iii) age (*1* and *3* days old unmated individuals: melanization assay; or *1, 3,* and *5* days old unmated individuals: bacterial survival and mosquito mortality) with the potential for both two and three-way interactions. *Model 2* explored the

effects of (i) convergence status, (ii) sex, and (iii) mating status in 3-day-old individuals only. Finally, *Model 3* explored the effects of (i) convergence status and (ii) blood-fed status and the potential for two-way interactions in 5-day-old females only. All effects were specified as categorical in nature with the exception of age, which was included as a continuous variable, but after centering and scaling with the means and standard deviations respectively[123]. Intercepts were allowed to vary randomly among parental pairs. Results of the melanization experiment were analyzed by ordered logistic regression with mixed effects using the package "ordinal"[124] The dependent variable was specified as non-, partially- or fully melanized beads in increasing order. For analyzing the in vivo bacterial growth assay, statistical models were specified with fluorescence intensity (after subtracting background fluorescence and averaging across duplicates) as the dependent variable. Linear mixed models were initially tested after transformation with $[\log(y + 1)]$ or without transformation of fluorescent intensities, but neither approach was able to account for the heteroscedasticity. However, rounding fluorescence intensities to the nearest integer allowed modeling the variable as a discrete count instead of using generalized linear mixed models (GLMMs), as described for another dataset[123]. All GLMMs were performed in the "glmmTMB" package[123] and specified with a Generalized Poisson distribution and "log" link. Intercepts were allowed to vary randomly among parental pairs across all model analyses. Finally, to analyze mosquito mortality following bacterial challenge, linear mixed models were specified as suggested in the package "lme4"[125]. The number of dead mosquitoes at each time point was specified as the dependent variable with the total number of mosquitoes in each cup included as an offset to account for differences between cups.

For analyzing DENV vector competence, binomial GLMMs with a logit link were performed, also in the "glmmTMB" package. The dependent variable was expressed as the proportion of mosquitoes positive for DENV-2 from the total number sampled. Fixed effects consisted of (1) convergence status (*non-converged* vs. *converged*), (2) tissue (*body*, *head and legs*, or *saliva*), and (3) time course of infection (days post-infection) modeled up to four-way interactions. While treatment and tissue were specified as categorical predictors, time post-infection was centered and scaled as described above. Further, to estimate how the probability of mosquitoes becoming infected, disseminating infection, or becoming infectious changed over the post-infection period, time was specified as both linear and quadratic ("hump-shaped") effects. Since multiple tissues were sampled for DENV from the same individual, random intercepts for mosquito number were also nested within each biological replicate. To further investigate whether the overall temporal trends in the proportion of mosquitoes that became infected, disseminated infection, or became infectious over time differed between converged and non-converged groups we ran an additional, separate model analysis for offspring from converged and non-converged parents. These models used the same fixed and random effect structure as described above with the exception that time (days post-infection) was specified as a categorical/group-level predictor.

For all model analyses, model selection and choice of the family were based on likelihood-based information criteria assessed using the package "bbmle"[126]. Reference groups for all analyses were chosen as follows: non-converged for convergence status, males for sex, not mated for mating status, blood-fed for blood-feeding status. For all the models, residuals were examined for normality with the "DHARMa" package[127] using a predetermined threshold ratio of squared Pearson residuals over the residual degrees of freedom <1.5 and a Chi-squared distribution of the squared Pearson residuals with $p > 0.05$. Once overdispersion was accounted for, the marginal means estimated by the model were used to perform pairwise comparisons using Tukey's contrasts in the "emmeans" package[128]. Statistical analysis for mosquito survival after DENV infections was performed using Kaplan–Meyer survival analysis (log-rank test).

**Reporting summary**. Further information on research design is available in the Nature Research Reporting Summary linked to this article.

## Data availability
The datasets generated and analyzed during the current study are available in the Dryad repository (https://doi.org/10.5061/dryad.ffbg79ct0).

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

## Author contributions

CCM, LJC, and LCH conceived of the study, CMR and CCM designed the experiments, CMR performed and analyzed the experiments and data, AKP derived the models and analyzed the data. CMR and CCM wrote the manuscript in consultation with LJC and CCM, LCH. MAB provided cell lines, virus stocks, and training.

## Competing interests

The authors declare no competing interests.
