## [Transparent Peer Review File · Communications Biology]

Reviewers' comments:

Reviewer #1 (Remarks to the Author):

The manuscript by Reitmayer et al presents a complex study that assessed the effects of mosquito age, sex, blood feeding, and acoustic convergence on the melanization immune response and the ability to contain a bacterial (*E. coli*) or viral (Dengue) infection. The authors find that the response to infection is shaped by the interaction of several factors. In other words, the response is complex, and not predominantly driven by any one of the factors studied.

This study was a significant undertaking because of the number of treatments and variables, which required the analysis of a large number of mosquitoes. It truly is impressive. The data are quite valuable because they consider the complexity of conditions commonly faced in the environment, so although no one specific predictor of the phenotypes could be discovered, the data provide a more realistic explanation of how mosquitoes respond to infection in real life.

I do not have any significant comments or criticisms of this manuscript. Perhaps the most salient comment is that because of all the different treatments and comparisons, the reader can easily get lost in the mix and fail to grasp the major findings of the manuscript. That is, the message is a bit lost in the details.

Minor comments:

1. Some justification as to why the three major types of experiments were selected would improve clarity. For example, assessing the strength of the melanization response is important, but then the study continues with *E. coli*, which is a bacterium that is primarily killed via phagocytosis. Moreover, why did one experiment measure a direct immunity phenotype (melanization) but the others focused on more indirect measurements of the consequence of infection (intensity and survival)?
2. It was curious that only one group reduced *E. coli* infection intensity: the blood fed group. Consider that blood feeding induces hemocyte mitosis, which likely increases the phagocytic response against *E. coli*.

Reviewer #2 (Remarks to the Author):

In this manuscript, Reitmayer et al provide results from several intricate mating, microbial challenges and DENV infection to deduce the effects of harmonic convergence signals on immune competency of offspring. When describing how harmonic convergence affects melanization response, the effect was only seen in males and not females. Given that females will be more affected by diminished melanization response as they encounter human pathogen via blood feeding, it is not clear what impact does this finding have with regards to overall vector competence.

When looking at the effects of harmonic convergence on bacterial challenge, it seems that it is visible on 1d old mosquitoes and less so at 3d and 5d. It doesn't seem that there isn't any strong effect on female ability to limit bacterial infection.

Furthermore, the results show that harmonic convergence does not affect overall DENV infection, disseminated infections and infectious status in female offspring. It only affected early infection dynamics. It is not clear what impact does it have with the overall mosquito vector competence. An intriguing result is that presented in fig3.B, where a clearer distinction is seen between converged and not converged offspring groups. However, there is no additional information on the immune status of these mosquitoes at this age, from where we could glean and associate any refractoriness or susceptibility with enhanced/diminished expression of key anti-DENV effectors. This substantially limits what we can extrapolate from this study with regards to immunity.

This manuscript is well written and has a well-organized and logic flow. It also describes interesting findings. However, after reading the title and each experimental segment, I was hoping to see more results/information on the mosquito immune status from these harmonic convergence groups that might support the author's hypothesis and explanation of their findings. I am left wondering what additional defense systems could be implicated in the phenotype they were observing. Given that we have ample knowledge on the antimicrobial repertoire against bacteria and DENV, it would have been appropriate to dig a bit deeper in assessing these immune pathways/mechanisms. In this regard, this manuscript would have benefited from a much more detailed examination of the mosquito immune system via qPCR of selected key marker genes, to further discern how acoustic mating interactions affect the immune system of *Aedes* offspring.

Having said that, I commend the authors for their effort in conducting these studies, since they require a lot of work and dedication to obtain the data.

Minor comments:

Line 169-170 "... indicates that overall virulence of bacterial infection increases with age more so for male than female offspring". Virulence is a bacterial attribute that should not change as you are using the same bacterial species, what changes is the susceptibility of your host. So remove virulence and state "... bacterial infection increases with age..."

Line 309-310 What do you mean by "better body condition"? do you mean immune competent? Physiologically "normal"? please define or change the wording.

The results/main messages might benefit from a pictogram describing the interactions observed in this manuscript.

Although described early on the results page, it would be good to mention how many replicates and individuals within each replicate were conducted for each of the experiments.

Graphs needs to have a x-axis title and statistical significance has to be much more explicitly shown on the figure or described in the figure legends (i.e. Fig1 A, what significance levels do the male vs females (lines) have?; Fig 1.B, here should Table S2B (page 2 Supplemental material) be Table S1B ? Add graph legends to each graph. It is much better appreciated than having the reader look for the graph legend description in the figure description. The figure has to stand alone and give as much information as possible.

Response to Reviewers' Comments:

We would like to thank the reviewers for all of their thoughtful and insightful comments. After incorporating the below suggestions, we think the manuscript has been improved greatly. We hope the revisions meet the reviewers' expectations. Reviewer comments are included in *italics*, with our responses in normal text. Changed text in the manuscript have been highlighted using track changes in the new version of the manuscript.

Thank you!

Courtney Murdock

Reviewer #1:

The manuscript by Reitmayer et al presents a complex study that assessed the effects of mosquito age, sex, blood feeding, and acoustic convergence on the melanization immune response and the ability to contain a bacterial (E. coli) or viral (Dengue) infection. The authors find that the response to infection is shaped by the interaction of several factors. In other words, the response is complex, and not predominantly driven by any one of the factors studied.

This study was a significant undertaking because of the number of treatments and variables, which required the analysis of a large number of mosquitoes. It truly is impressive. The data are quite valuable because they consider the complexity of conditions commonly faced in the environment, so although no one specific predictor of the phenotypes could be discovered, the data provide a more realistic explanation of how mosquitoes respond to infection in real life.

I do not have any significant comments or criticisms of this manuscript. Perhaps the most salient comment is that because of all the different treatments and comparisons, the reader can easily get lost in the mix and fail to grasp the major findings of the manuscript. That is, the message is a bit lost in the details.

We thank reviewer #1 for the positive comments on our manuscript! To clarify for the reader why we focused on assessing immune performance across the various treatments in addition to harmonic convergence (sex, age, physiological status, and mating status), we built in text into the introduction (Lines 116-122) and at various points throughout the discussion.

Minor comments:

1. Some justification as to why the three major types of experiments were selected would improve clarity. For example, assessing the strength of the melanization response is important, but then the study continues with E. coli, which is a bacterium that is primarily killed via phagocytosis. Moreover, why did one experiment measure a direct immunity phenotype (melanization) but the others focused on more indirect measurements of the consequence of infection (intensity and survival)?

The reviewer brings up a good point. We choose our particular assays for a variety of reasons. Since we do not know which parasites / pathogens are imposing selective pressures on mosquito immunity, we wanted to focus on immune phenotypes that comprise multiple

arms of immune defense (humoral, melanization; and cellular, bacterial killing), are important for a wide diversity of parasites and pathogens, and are energetically costly. Further, these immune assays are well established approaches and have been used widely for mosquitoes in previous studies and will allow us to assess effects of convergence, sex, age, etc. on whole immune phenotypes that will guide future mechanistic work. Finally, because female *Ae. aegypti* are important vectors for human parasites / pathogens, we also included a relevant pathogenic challenge to assess if harmonic convergence had implications for transmission. Justification for each has been incorporated in the Introduction (Lines 111-116) and Discussion (Lines 225-228, Lines 245-251, Lines 280-283).

2. It was curious that only one group reduced E. coli infection intensity: the blood fed group. Consider that blood feeding induces hemocyte mitosis, which likely increases the phagocytic response against E. coli.

Thank you for catching this. We had forgotten to address this in the discussion. We have now added this insight as well as supporting references in the Discussion (Lines 257-258).

Reviewer #2:

In this manuscript, Reitmayer et al provide results from several intricate mating, microbial challenges and DENV infection to deduce the effects of harmonic convergence signals on immune competency of offspring. When describing how harmonic convergence affects melanization response, the effect was only seen in males and not females. Given that females will be more affected by diminished melanization response as they encounter human pathogen via blood feeding, it is not clear what impact does this finding has with regards to overall vector competence.

Thank you for your comment. To clarify, the objective of this study was not solely to address how harmonic convergence shapes the vector competence of females for pathogens / parasites of human concern, but more broadly to explore how harmonic convergence affects the immune performance of both male and female offspring. Our lack of knowledge on the selective pressures that shape mosquito immunity in nature is a critical gap in our understanding of mosquito-pathogen interactions. To date, much of the mosquito immune response has solely focused on females of a certain age, under highly controlled laboratory conditions, and on immune defenses that are important for human pathogens. Our study demonstrates that our understanding of the mosquito immune system is incomplete, with mosquito immunity being a phenotype that changes dynamically with mosquito age, physiological state, and mating status. Further, female immunity does not necessarily translate to male immunity, with the sexes facing different life history constraints. This is especially relevant for novel mosquito control technologies (e.g. wolbachia infections, transgenic techniques, etc.) that manipulate mosquito immunity. These technologies could indirectly impact the mating competitiveness of male mosquitoes via physiological trade-offs between mating and immune investment, and the efficacy of these technologies could vary as mosquitoes age and at different life history stages.

When looking at the effects of harmonic convergence on bacterial challenge, it seems that it is visible on 1d old mosquitoes and less so at 3d and 5d. It doesn't seem that there isn't any strong effect on female ability to limit bacterial infection. Furthermore, the results show that harmonic convergence does not affect overall DENV infection, disseminated infections and infectious status in female offspring. It only affected early infection dynamics. It is not clear

what impact does it have with the overall mosquito vector competence. An intriguing result is that presented in fig3.B, where a clearer distinction is seen between converged and not converged offspring groups. However, there is no additional information on the immune status of these mosquitoes at this age, from where we could glean and associate any refractoriness or susceptibility with enhanced/diminished expression of key anti-DENV effectors. This substantially limits what we can extrapolate from this study with regards to immunity.

We agree with the reviewer's points that at this stage, it is difficult to discern what mechanisms might be influencing the dynamics and overall effects of harmonic convergence on the proportion of mosquitoes that become infected, disseminate infection, and become infectious (e.g. metrics of vector competence). The objective of this study was to first establish how harmonic convergence impacts immune phenotypes (whole immune responses or outcomes of infection). We feel that this step is important to complete first, as it facilitates more informed and targeted hypotheses and downstream experiments addressing possible mechanisms. As such, we had outlined what we think are potential hypotheses that might explain the differences in DENV infection dynamics between females from harmonically converged or not converged parents throughout the Discussion and plan to address these in future work.

This manuscript is well written and has a well-organized and logic flow. It also describes interesting findings. However, after reading the title and each experimental segment, I was hoping to see more results/information on the mosquito immune status from these harmonic convergence groups that might support the author's hypothesis and explanation of their findings. I am left wondering what additional defense systems could be implicated in the phenotype they were observing. Given that we have ample knowledge on the antimicrobial repertoire against bacteria and DENV, it would have been appropriate to dig a bit deeper in assessing these immune pathways/mechanisms. In this regard, this manuscript would have benefited from a much more detailed examination of the mosquito immune system via qPCR of selected key marker genes, to further discern how acoustic mating interactions affect the immune system of Aedes offspring.

Having said that, I commend the authors for their effort in conducting these studies, since they require a lot of work and dedication to obtain the data.

We agree that it would be interesting to look at immune pathway parameters known to impact the mosquito anti-bacterial and anti-viral response. Please see our response to the previous comment. Thank you for your positive remarks on the manuscript!

Minor comments:

Line 169-170 "... indicates that overall virulence of bacterial infection increases with age more so for male than female offspring". Virulence is a bacterial attribute that should not change as you are using the same bacterial species, what changes is the susceptibility of your host. So remove virulence and state "... bacterial infection increases with age..."

Changes made according to reviewer's suggestions (Lines 164-165)

Line 309-310 What do you mean by "better body condition"? do you mean immune competent? Physiologically "normal"? please define or change the wording.

Changes made according to reviewer's suggestions Lines 308-309.

The results/main messages might benefit from a pictogram describing the interactions observed in this manuscript.

We are a bit unclear what the reviewer is suggesting here. Are you referring to a general schematic on our findings that highlights the effects of harmonic convergence on our response variables for each assay? If so, and if the editor agrees, we could try and put something together on this front. We do feel that this would have to be done carefully so as not to be repetitive from our current figures, which visualize the effects harmonic convergence, sex, age, physiological status, and mating status on our response variables.

Although described early on the results page, it would be good to mention how many replicates and individuals within each replicate were conducted for each of the experiments.

Currently this information is presented in the Results (Lines 125-136). Further, the number of parental pairs (replication) and sample sizes are also included in each of the summary analysis tables in the supplementary information.

Graphs needs to have a x-axis title and statistical significance has to be much more explicitly shown on the figure or described in the figure legends (i.e. Fig1 A, what significance levels do the male vs females (lines) have?;

We are not sure which graphs reviewer #2 is referring to as each graph already shows an x-axis title. Unfortunately, due to the complexity of the statistical models and the number of interactions investigated it is not possible to state each interaction/main effect significance in a standard figure legend. When possible (with lower order interactions), we have included labeling to denote the significant pair-wise comparisons. However, for the higher order interactions where this would be intractable, we ask the reader to refer to the related SI pair-wise comparisons of interest and the significance values associated with those comparisons.

Fig 1.B, here should Table S2B (page 2 Supplemental material) be Table S1B ?

Thank you, these have now been corrected.

Add graph legends to each graph. It is much better appreciated than having the reader look for the graph legend description in the figure description. The figure has to stand alone and give as much information as possible.

We are not sure what reviewer 2 means here, but to improve overall clarity of the figures we have remade them. See new figures 1-3 and associated figure legends.